# Rendezvous at Plasma Membrane: Cellular Lipids and tRNA Set up Sites of HIV-1 Particle Assembly and Incorporation of Host Transmembrane Proteins

**DOI:** 10.3390/v12080842

**Published:** 2020-07-31

**Authors:** Dishari Thornhill, Tomoyuki Murakami, Akira Ono

**Affiliations:** 1Department of Microbiology and Immunology, University of Michigan, 1150 West Medical Center Drive, 5737 Medical Science II, Ann Arbor, MI 48109-5620, USA; dishari@umich.edu; 2University of Michigan, 1150 West Medical Center Drive, 5737 Medical Science II, Ann Arbor, MI 48109-5620, USA; tmurakam@umich.edu

**Keywords:** HIV-1, PI(4,5)P2, cholesterol, RNA, tetherin, CD44, CD43, PSGL-1, assembly, membrane microdomains

## Abstract

The HIV-1 structural polyprotein Gag drives the virus particle assembly specifically at the plasma membrane (PM). During this process, the nascent virion incorporates specific subsets of cellular lipids and host membrane proteins, in addition to viral glycoproteins and viral genomic RNA. Gag binding to the PM is regulated by cellular factors, including PM-specific phospholipid PI(4,5)P2 and tRNAs, both of which bind the highly basic region in the matrix domain of Gag. In this article, we review our current understanding of the roles played by cellular lipids and tRNAs in specific localization of HIV-1 Gag to the PM. Furthermore, we examine the effects of PM-bound Gag on the organization of the PM bilayer and discuss how the reorganization of the PM at the virus assembly site potentially contributes to the enrichment of host transmembrane proteins in the HIV-1 particle. Since some of these host transmembrane proteins alter release, attachment, or infectivity of the nascent virions, the mechanism of Gag targeting to the PM and the nature of virus assembly sites have major implications in virus spread.

## 1. Introduction

Successful assembly of infectious virus particles is an essential step in virus spread from infected to uninfected cells. For HIV-1, the assembly process takes place at the plasma membrane (PM) in most cells, including the physiological host cell type, CD4^+^ T cells [1,2]. This process is driven by the viral structural protein Gag, synthesized as a precursor polyprotein Pr55Gag and comprising the following steps: 1) targeting of Gag to the site of virus assembly, 2) binding of Gag to the lipid bilayer, 3) recruitment and packaging of viral genomic RNA, 4) multimerization of Gag, 5) incorporation of viral Env glycoproteins, and 6) budding and release of nascent virus particles (not necessarily in this order for steps 1 to 5).

The HIV-1 Gag matrix (MA) domain mediates the localization and binding of Gag to the PM [3,4]. In CD4^+^ T cells that have a polarized morphology, Gag localizes to the PM of a rear-end protrusion known as uropod [5,6,7,8]. For Gag localization to this specific region of the PM, NC-dependent Gag multimerization is necessary, in addition to MA-mediated PM binding [5].

The assembling virus incorporates not only viral proteins and RNAs but also host PM components, i.e., lipids and host-encoded membrane proteins present at the virus assembly sites. These host proteins could modulate the production, transmission, and/or entry of progeny virions [9,10,11,12]. Therefore, further understanding of how these components are incorporated into the virus particles can help inform antiviral strategies. Notably, when HIV-1 Gag mislocalizes to intracellular membranes, virus-like particles are formed at these internal membrane sites [1,13,14,15,16,17,18,19]. Thus, subcellular Gag localization plays a major role in determining host membrane components incorporated into virions.

In this review, we will compile and examine our current understanding of the mechanism that directs Gag to the PM and the membrane environment at the site of virus assembly. We will then explore how the virus assembly site at the PM may influence the selection of host proteins into the HIV-1 particle. The major points covered in this review will be:the interplay between PI(4,5)P2 and tRNA in mediating specific localization of Gag to the PM,the effects of PM lipids on HIV-1 assembly, andthe potential roles played by the membrane environment in the incorporation of a subset of transmembrane proteins, which localize to uropods in T cells.

## 2. The Roles Played by PI(4,5)P2 in Specific Localization of Gag

Specific localization of Gag to the PM is mediated by the MA domain [1,2,4]. MA employs a bipartite signal to stably interact with lipid bilayers; an N-terminal myristate moiety and a highly basic region within its globular domain (MA-HBR) [13,15,18,20,21,22,23]. The myristate moiety is a 14-carbon fatty acid co-translationally attached to the N-terminal glycine of MA [24]. When the myristate moiety is not sequestered within the MA globular head, it mediates nonspecific hydrophobic interactions of Gag with membranes [25,26,27,28,29,30,31,32]. By itself, the myristate moiety is not sufficient to stably anchor Gag to the lipid bilayer [24]. The second determinant for the membrane binding of HIV-1 Gag is the MA-HBR (Figure 1a), a conserved patch of basic amino acids encompassing MA residues 14–31, which interacts with negatively charged lipids [23,33,34]. Earlier in vitro studies showed that HIV-1 MA can bind lipid membranes containing acidic phospholipids such as phosphatidylserine (PS) [23,35,36]. However, as described below, multiple lines of evidence indicate that MA-HBR preferentially binds to an anionic lipid that is enriched at the PM, phosphatidylinositol-(4,5)-bisphosphate [PI(4,5)P2], and that this interaction promotes specific binding of Gag to the PM [13,30,37,38,39,40,41,42,43,44,45,46,47].

PI(4,5)P2 is a polyphosphoinositide bearing a glycerol backbone, with two acyl chains esterified at the sn−1 and sn−2 positions and a phosphate and myo-inositol ring the sn−3 position (Figure 1b). The myo-inositol ring in PI(4,5)P2 is phosphorylated at the fourth and the fifth positions. PI(4,5)P2 is primarily present in the inner leaflet of the PM, with minor populations distributed at other subcellular compartments [53,54,55]. Phosphatidylinositol phosphate (PIP) kinases (PIPKs), which generate PI(4,5)P2 at specific membranes, can be divided into two groups: type I (PIPKIs) and type II (PIPKIIs). Type I phosphatidylinositol 4-phosphate 5-kinases (PI4P5KI, one of PIPKIs), which preferentially phosphorylate the 5-hydroxyl group on the inositol ring of PI(4)P to produce PI(4,5)P2, are the dominant PIPKs in mammalian cells. A relatively lesser pool of PI(4,5)P2 is produced by PIPKIIs, which use PI(5)P as substrate and phosphorylate PI(5)P at the 4-hydroxyl position. Various polyphosphoinositide kinases and phosphatases act in concert to cycle PIs between different phosphorylation states. PI(4,5)P2 can be dephosphorylated by a subset of polyphosphoinositide phosphatases. The polyphosphoinositide 5-phosphatase INPP5E, also known as 5-phosphatase IV, converts PI(4,5)P2 to PI(4)P by dephosphorylating at the 5-hydroxyl position of the inositol ring [56]. Thus, spatio-temporal distribution of PI(4,5)P2 is orchestrated by the presence of PIPKs, PI(4,5)P2 phosphatases, the enzyme substrates [PI(4)P and PI(5)P], as well as PI(4,5)P2 binding/effector proteins [49,54,55,57,58,59,60,61].

When PI(4,5)P2 is depleted from the PM by the overexpression of 5-phosphatase IV or when the localization of PI(4,5)P2 is altered by inducing the formation of PI(4,5)P2-laden endosomal vesicles, the PM localization and subsequent virus-like particle (VLP) release of HIV-1 Gag is severely decreased [13,38,39,43,45,48,62,63,64], emphasizing the importance of PI(4,5)P2 in HIV-1 assembly and release. In PI(4,5)P2-depleted cells, Gag localizes promiscuously to the cytoplasmic membranes or fails to bind to any membrane and remains cytosolic. Through these studies, PI(4,5)P2 was shown to be an important cellular cofactor for HIV-1 assembly and subcellular Gag localization in multiple cell types [39,64], including the physiological host T cells [62]. A study using an inducible PI(4,5)P2 depletion system, which is based on an engineered 5-phosphatase IV, showed that even Gag multimers detach from the PM upon PI(4,5)P2 depletion [63]. Other approaches to perturb PIPs have also been used to examine the role of PI(4,5)P2 in HIV-1 assembly. For example, the expression of a constitutively active Arf6, which induces a PI(4,5)P2-enriched endosomal compartment, was observed to relocalize Gag to the induced compartment [13]. Another study showed that the inhibition of Rab27-dependent trafficking of PI4KIIα, an enzyme that catalyzes the production of PM-specific PI4P, reduced PI(4,5)P2 at the PM and suppressed Gag localization to the PM in T cells, highlighting the complexity of the PI cycle regulating HIV-1 assembly [65]. In HeLa-derived cells, knockdown of PI4P5KIα and γ, but not β, impaired the targeting of Pr55Gag to the PM and caused mislocalization of Gag to intracellular compartments [66]. Therefore, it is conceivable that HIV-1 Gag binds to a specific pool of PI(4,5)P2 in cells. Perturbation of cellular PI(4,5)P2 or other phosphoinositides has been shown to reduce particle production of other retroviruses, although some retroviruses are less sensitive to 5-phosphatase IV expression [19,39,43,64,67,68,69,70,71].

## 3. Determinants for Gag Binding to PI(4,5)P2

The PI(4,5)P2 headgroup is more negatively charged than a ubiquitous acidic phospholipid PS (the net charge of PI(4,5)P2, −3 or −4; PS, −1 [72]). Supporting the charge-based preference of Gag for PI(4,5)P2, a modeling study on MA-membrane interactions predicted that nonspecific electrostatic interactions between MA and PI(4,5)P2 were sufficient to significantly enhance membrane binding [34]. However, in vitro studies showed that HIV-1 Gag or MA binds more efficiently to PI(4,5)P2-containing lipid membranes than to charge-matched liposomes containing PS, suggesting that PI(4,5)P2 enhances Gag-membrane binding by specific interaction beyond mere electrostatic attraction [38,41,44,45,73,74,75,76,77]. Membrane flotation-based studies showed that PI(3,5)P2 and PI(3,4,5)P3 were also able to enhance the binding of Gag to liposomes, although the former is not as efficient [38]. Likewise, an NMR-based analysis showed enhanced binding of MA to liposomes containing PI(3,5)P2 as well, although not as efficiently as PI(4,5)P2 [78]. Therefore, both negative charge density of the inositol headgroup and distribution of the phosphate residues on the inositol ring are likely determinants for the optimal interaction between MA and PI(4,5)P2. Recent genetic and structural studies collectively suggest that HBR residues Lys 29 and Lys 31, especially Lys 31, play key roles in the interaction with the PI(4,5)P2 headgroup, although other basic residues may also be involved [41,48,78,79,80]. In addition, as we discuss later, binding of RNA to MA-HBR residues is also likely to contribute to lipid specificity.

The sn-2 position of the glycerol backbone of PI(4,5)P2 is most commonly decorated with arachidonic acid. The 1-stearoyl-2-arachidonyl (38:4) form of PI(4,5)P2 is found to be a dominant species across multiple primary mammalian cells [49,50,51,52,53] (Figure 1b). The preference of enzymes involved in the PI cycle, such as diacylglycerol kinase ε (DGKε), for substrates displaying 38:4 fatty acyl chains likely explains the enrichment of 38:4 PI(4,5)P2 [51,57]. Although an earlier structural study using water-soluble PI(4,5)P2 derivatives with short acyl chains proposed that the sn-2 acyl chain interacts with a hydrophobic cleft of the MA globular domain [30], subsequent structural and in silico studies suggest that PI(4,5)P2 acyl chains with native lengths are unlikely to directly interact with MA [78,81]. Nonetheless, the acyl chains of PI(4,5)P2 may play an indirect role in MA binding. An in vitro study using giant unilamellar vesicles (GUVs) showed that Gag prefers PI(4,5)P2 with at least one unsaturated acyl chain [82]. In the same experimental system, the pleckstrin homology domain of phospholipase Cδ1 (PH_PLCδ1_), which interacts with the headgroup of PI(4,5)P2, did not show a bias towards specific PI(4,5)P2 acyl chains unlike Gag [82]. In the presence of cholesterol (discussed later) or a higher concentration of PS, HIV-1 Gag binding to liposomes can be detected even in the absence of PI(4,5)P2. Under these conditions, HIV-1 Gag shows a preference for PS with unsaturated acyl chains over PS with saturated acyl chains [83], whereas RSV Gag, which lacks N-terminal myristoyl moiety, does not show such preference [84,85]. It will be intriguing to examine whether the myristoyl moiety plays a role in the observed acyl chain preference, for example, by favoring a loose packing (warranted by unsaturated acyl chains) around the acidic phospholipid that Gag binds. Lipidomics studies showed that unsaturated acyl chain species, including 38:4, were the major PI(4,5)P2 acyl chain species found in the HIV-1 particle [39,86], but this can be largely accounted for by the abundance in the PM. Currently, whether the unsaturation of acyl chains is necessary for Gag binding to PI(4,5)P2 in cells remain unclear.

## 4. The Balance Between MA Binding to PI(4,5)P2 and RNA

The MA-HBR has been shown to bind not only acidic lipids but also RNA [48,87,88,89,90,91,92]. Notably, in experimental systems where full-length Gag is translated in vitro using eukaryotic cell lysates and examined for binding to liposomes containing PS, RNA that is present in the cell lysates acts as a negative regulator for Gag membrane binding. Only when RNA is removed by treatment with RNase or when PI(4,5)P2 is included in the liposomes, in vitro transcribed Gag binds liposomes efficiently [38,41,45,82]. Thus, RNA bound to Gag via MA inhibits Gag binding to PS, but PI(4,5)P2 is able to overcome the negative regulation by RNA (Figure 2a).

The in vitro observations of competition between RNA/nucleotide and acidic lipids [41,46] led to a proposal in which RNA binding to MA prevents MA interaction with PS, an acidic phospholipid ubiquitously distributed in cells, and thereby ensures specific binding of Gag to the PM, which contains PI(4,5)P2 (Figure 2a). In support of this model, Gag binding to RNA via MA in cells has been observed [48,89,93]. Moreover, RNA bound to MA in cells suppresses the membrane binding of Gag molecules present in the cytosol of HIV-expressing cells regardless of whether the membrane is PS-containing liposomes [93] or total cellular membranes [89]. But does RNA binding to MA prevent Gag from mislocalizing to the intracellular membranes containing PS in cells? Our comparison of HIV-1 Gag chimeras containing MA domains of various retroviruses showed that MA domains sensitive to RNA-mediated inhibition of Gag-liposome binding direct Gag chimeras to the PM, whereas Gag chimeras containing MA domains that are insensitive to RNA-mediated inhibition show promiscuous subcellular localization [43,45]. Furthermore, a strong correlation between MA-RNA binding in cells and the prevention of Gag mislocalization has been observed in our analysis of HIV-1 MA mutants [48]. This study compared Gag derivatives in which two of the eight basic amino acids in MA-HBR are substituted from Lys to Arg (KR) (preserving the total positive charge of MA-HBR) or from Lys to Thr (KT) (reducing the total positive charge) (Figure 1a). Consistent with the importance of the HBR charge in RNA binding, the KT mutants failed to bind RNA via MA efficiently in cells, unlike the KR mutants. These KT mutants bound both PM and intracellular membranes indiscriminately. In contrast, one KR mutant bound PM exclusively, consistent with the role for total MA-HBR positive charge and/or MA-bound RNA in navigating Gag specifically to the PM. Another KR mutant was predominantly cytosolic, despite retaining the total positive charge; however, when its membrane binding was enhanced through augmented Gag multimerization, it achieved specific binding to the PM without localization to other compartments. Altogether, these studies provide cell-based data supporting a model that MA binding by RNA prevents mislocalization of Gag in cells, and along with PI(4,5)P2, helps Gag localize specifically to the PM. Thus, it appears that a fine balance between MA-PI(4,5)P2 and MA-RNA binding exists and dictates the subcellular localization of Gag.

The majority of RNA that binds MA-HBR in cells has been identified as transfer RNA (tRNA) [89]. Interestingly, the affinity of MA binding to tRNA [90,94] surpasses its affinity to PI(4,5)P2 [73,78,79]. The affinity of MA to PI(4,5)P2 can be augmented when PS or cholesterol is included in the PI(4,5)P2-containing membranes [73], when MA is artificially trimerized [79], or when MA is myristoylated [73]. However, there has been no side-by-side comparison of MA binding affinity to membranes versus tRNA using identical experimental conditions and methods thus far. Of note, in the in vitro liposome binding assays in which in vitro transcribed full-length Gag is examined for liposome binding, RNase treatment drastically increases Gag binding to liposomes containing PS, but it also increases Gag binding to PI(4,5)P2-containing liposomes [41]. This observation suggests that Gag binding to PI(4,5)P2 can be susceptible to the RNA-mediated inhibition depending on the condition. Consistent with this notion, while in one in vitro study, tRNA below typical cellular concentrations suppressed binding of Gag to PC + PS but not PI(4,5)P2-containing membranes [93], in other in vitro systems where purified MA or Gag are added to lipid membranes, the addition of tRNAs at similar tRNA concentrations reduced membrane binding of MA/Gag to various degrees even when the membranes contain PI(4,5)P2 [74,76,90]. Therefore, the balance between MA binding to PI(4,5)P2 and tRNA can be affected by factors that differ among experimental systems. These factors may include the state of Gag multimerization, the presence of other lipids, the effects of divalent cations on tRNAs and membranes, and the clustering of PI(4,5)P2 (see below). Apart from the differences among in vitro systems, there are several caveats that exist for in vitro studies. One is that most tRNAs in cells may be in association with the cellular translational machinery and not free, thus the actual concentration of free tRNA in cells is difficult to assess. In addition, tRNAs shuttle between different cellular compartments in response to cellular stress [95,96], and therefore, the concentrations of tRNA in the cytoplasm may change during productive infection. Furthermore, in vitro transcribed tRNAs, which were used in some of the in vitro Gag/MA membrane binding studies, lack posttranscriptional modifications which are important for tRNA folding, structure and stability in the cell [97,98,99,100,101]. Therefore, further validation of the in vitro results regarding MA-tRNA and MA-PI(4,5)P2 associations with cell-based systems is warranted.

MA-RNA binding is mediated by the MA region containing HBR [37,48,87,88,89,90,93,102,103,104,105,106,107,108]. Therefore, it is likely that tRNA binding suppresses Gag membrane binding through inhibition of the interaction between HBR and acidic phospholipids. Interestingly, however, our previous study has shown that removal of RNA from Gag by RNase treatment increases not only Gag binding to negatively charged liposomes but also, to a lesser extent, binding to liposomes containing only a neutral lipid PC [41]. Therefore, hydrophobic interactions mediated by the myristoyl moiety may also be regulated by RNA binding to MA-HBR. A similar observation was also made in another in vitro study that used a purified Gag and yeast tRNA [76]. Of note, myristoylated MA under conditions that favor the myristate exposure has a weakened affinity to tRNA compared to when the myristate moiety is sequestered [90]. Therefore, there may be an interplay between myristoyl exposure and MA-tRNA binding. A potential mechanism may be that tRNA binding to specific MA-HBR residues prevents the exposure of the myristate moiety. However, conditions that trigger the exposure of the myristate moiety, such as Gag multimerization or a chance apposition to lipid bilayers, or the exposed myristate moiety itself may weaken tRNA binding. Obviously, these speculations need to be tested experimentally.

## 5. Enrichment of PI(4,5)P2 in HIV-1 Particles

The estimated amounts of PI(4,5)P2 vary depending on the type of cells. PI(4,5)P2 resides mainly in the inner leaflet of the PM at ~1–2 mol% with ~5000–20,000 molecules of PI(4,5)P2 per μm^2^ of PM [57]. PI(4,5)P2 is found to be enriched in the HIV-1 envelope as compared to the PM of the producer cell [39,86,109,110], and this enrichment is dependent on MA [39]. There are an estimated 8000 molecules of PI(4,5)P2 in an HIV-1 particle [86]. Total surface area for one HIV-1 particle will be 0.07 um^2^ assuming an average diameter of 150 nm for each virus particle. Thus, there is a ~5.6 to 22.4-fold increase in PI(4,5)P2 density on HIV-1 compared to that on the PM. A new nanodisc-based study determined that ~one PI(4,5)P2 molecule recruits one MA molecule (6 molecules of PI(4,5)P2 versus 5 molecules of MA per leaflet of the nanodisc) [79]. However, assuming an average of 2500 Gag molecules per virus particle, the ratio of PI(4,5)P2 to Gag is 3:1 in a virus particle as per the latest HIV-1 lipidomics study [86]. How does HIV-1 achieve such an enrichment of PI(4,5)P2 in its envelope?

The PM is not a homogeneous sea of lipids and proteins. Instead, the PM is a dynamic mosaic of various nanoscale or mesoscale domains (herein collectively referred to as microdomains) that have distinct lipid and protein signatures [111,112,113,114,115,116]. As for PI(4,5)P2, super-resolution microscopy studies in PC12 cells showed that PI(4,5)P2 exists in ~65–73 nm diameter clusters in the inner leaflet of the PM [117,118]. In model membranes, PI(4,5)P2 cluster formation can be driven by multivalent metal ions [119,120]. Metal ions generate PI(4,5)P2 clusters even at low PI(4,5)P2 concentrations of 0.02 mol% [120]. One can speculate that given that the MA-HBR is the established interface for the PI(4,5)P2-Gag association, the increase in PI(4,5)P2:Gag ratio in virus particle compared to nanodisc may be a result of PI(4,5)P2 clustering. In fact, a previous modeling study showed that a single basic peptide could induce the clustering of PI(4,5)P2 [121]. A combination of PI(4,5)P2 clustering and Gag multimerization is likely to lead to the incorporation of larger numbers of PI(4,5)P2 into the virus particle than what initially engaged Gag.

## 6. Roles Played by Cholesterol and Cholesterol-Rich Membrane Domains in HIV-1 Assembly

Another microdomain of note in HIV-1 assembly is the lipid raft, as reviewed previously [122]. Membrane rafts are highly dynamic, potentially heterogenous nanoscopic domains (10–200 nm) enriched in sterols and sphingolipids [123,124]. Segregation of these raft lipids from surrounding glycerolipid-rich regions is often correlated with the phase separation of lipid bilayers into rigid liquid-ordered (Lo) membrane and the adjacent highly fluid liquid-disordered (Ld) region [125,126]. Proteins modified with saturated acyl chains such as the palmitate moiety commonly partition into the Lo phase, which is thought to represent lipid rafts, while proteins bearing unsaturated acyl chains such as geranylgeranyl moiety do not [127,128,129,130].

Interestingly, depletion of cellular cholesterol, which disrupts lipid rafts, reduces Gag membrane binding in cells [131]. Additionally, in vitro studies have shown that cholesterol strengthens the association of Gag to liposomes in a concentration-dependent manner both in the absence and presence of PI(4,5)P2 [77,83]. Likewise, SPR on sparsely tethered bilayer lipid membranes (stBLMs) demonstrated that the inclusion of cholesterol increases the amount of myristoylated MA bound to stBLM [73]. Furthermore, an NMR-based study found that myristoylated MA exhibits significant affinity to raft-like lipids even in the presence of PI(4,5)P2 [78]. These studies strongly support the role of cholesterol in Gag membrane binding (Figure 2b).

## 7. Relationships Between PI(4,5)P2 and Cholesterol During HIV-1 Assembly

A substantial number of studies suggest that HIV-1 assembly takes place in the area of the PM enriched in lipid raft microdomains. The HIV-1 envelope is enriched with both cholesterol and sphingomyelin compared to producer cell membranes [39,86,109,110,132]. Consistent with this, cholesterol was apparently enriched at Gag assembly sites on the HeLa cell surface [133]. Moreover, the level of membrane order of viral envelope determined using Laurdan staining was within the range expected for Lo membranes [134]. Raft-preferring proteins are found enriched in the HIV-1 envelope [6,122,135,136,137,138]. Gag is recovered in detergent-resistant membrane (DRM) fractions [6,122,139,140,141,142,143,144,145], which are thought to contain lipid rafts, albeit DRM is unlikely to represent intact raft microdomains in cells [122,123,124]. Gag colocalizes with lipid raft markers in cells using immunofluorescence microscopy [6,122,144]. Depletion of cholesterol and/or sphingolipids in cells adversely affects normal Gag behavior in cells, virus particle production and/or infectivity [109,131,145,146,147]. Overall, the data support the role of cholesterol-enriched raft-like microdomains in HIV-1 particle production. Interestingly, however, in an in vitro study, an MA-based construct that binds PI(4,5)P2-containing GUV membrane upon induced multimerization is excluded from the Lo phase in model membranes. Instead, it colocalizes with Ld markers along with PI(4,5)P2 [75]. Thus, it is unlikely that Gag merely targets or buds through the existing raft-like domains to form virus particles, especially given the role of PI(4,5)P2 in recruiting Gag to the PM.

As discussed earlier, the majority of PI(4,5)P2 molecules in cells have an unsaturated acyl chain at its sn-2 position and are therefore less likely to partition into lipid rafts consisting of saturated lipids. This poses a question as to how HIV-1 particles can be enriched in PI(4,5)P2 and lipid raft components at the same time. It is conceivable that Gag multimerization on the membrane reorganizes the lipid landscape at the assembly sites. Consistent with this possibility, previous microscopy-based studies showed that Gag promotes colocalization of markers of lipid rafts and another distinct microdomain, tetraspanin-enriched microdomains (TEM) [148] at Gag assembly and budding sites [149,150]. Thus, Gag multimerization at the PM could lead to spatiotemporal rearrangement of membrane domains at the assembly sites, thereby allowing the distinctive HIV-1 envelope composition. Indeed, another PI(4,5)P2-binding protein, Annexin A2 (AnxA2), was found to induce the formation of microdomains rich in PI(4,5)P2, cholesterol, glycosphingolipids [151] at least in model membranes, supporting the possibility of protein-induced rearrangement of membrane microdomains.

In this regard, it is noteworthy that cholesterol facilitates the clustering of PI(4,5)P2 in model membranes in the presence [120] or the absence of multivalent cations [152]. Thus, the presence of cholesterol and Gag multimers, which are likely to bind multiple PI(4,5)P2, could prompt the formation and growth of PI(4,5)P2 nanodomains in the PM. In support of this possibility, Gag assembly on model membranes was shown to induce PI(4,5)P2 cluster formation [153]. In complex lipid mixtures, cholesterol but not sphingomyelin appeared to be co-clustered along with PI(4,5)P2 by Gag assembly [153]. Furthermore, HIV-1 Gag was found to restrict the mobility of both PI(4,5)P2 and cholesterol, but not sphingomyelin, in the PM of living CD4+ T lymphocytes by super-resolution microscopy with scanning fluorescence correlation spectroscopy [154]. Whether the microdomain reorganization mediated by Gag multimerization takes the shape of a new microdomain with an amalgamation of the raft- and nonraft-like components or whether there is a juxtaposition of two microdomains at submicroscopic scale will be interesting to determine.

## 8. Potential Roles Played by Membrane Environment at Virus Assembly Sites in Host Protein Incorporation into Virions

Membrane microdomain reorganization is likely to contribute to the enrichment of specific membrane proteins at the virus assembly sites. Indeed, host proteins that prefer the cholesterol-enriched raft-like microdomain cluster at the HIV-1 assembly sites [133,149] (Figure 3). However, a raft-like environment is not the sole determinant for protein recruitment to the HIV-1 assembly sites. For example, the association of the tetraspanin CD81 and some GPI-anchored proteins with the HIV assembly sites, as determined by copatching and correlative light and electron microscopy-based approaches, diminishes when Gag fails to form budding structures at the PM [149]. Furthermore, we and others have shown using super-resolution localization microscopy or live TIRF imaging that membrane curvature promotes the incorporation of tetherin to the virus assembly sites [133,155] (Figure 3a). Importantly, tetherin-mediated inhibition of virus release, which requires the incorporation of tetherin to nascent particles, is unaffected by cholesterol depletion of virus-producing cells [155]. Therefore, while tetherin possesses a GPI anchor, which is a canonical raft-targeting signal (Figure 3b), cholesterol-enriched microdomains do not appear to play a role in tetherin recruitment to nascent virions.

It is conceivable that PI(4,5)P2, another lipid that is enriched at virus assembly sites, also promotes the incorporation of specific host membrane proteins. We have found that a subset of host transmembrane proteins that localize to a T cell uropod, the rear-end protrusion of polarized immune cells, copatch with each other [5] and are specifically recruited to the HIV-1 assembly sites [156]. Interestingly, coclustering between Gag and these uropod-directed proteins (PSGL-1, CD43, and CD44), determined by super-resolution localization microscopy, is dependent on the MA-HBR [156]. Gag derivatives that are engineered to bind the PM without MA-HBR failed to cocluster with PSGL-1. Moreover, the coclustering with Gag requires juxtamembrane polybasic sequences in the cytoplasmic tails of PSGL-1, CD43, and CD44. These results are consistent with the possibility that Gag multimerization facilitates the formation of microdomains enriched in acidic lipids, in particular PI(4,5)P2, which in turn recruit PSGL-1, CD43, and CD44 via electrostatic interactions (Figure 3c). In support of this possibility, a structural study suggests that the CD44 cytoplasmic tail coclusters with PI(4,5)P2 [157]. However, PSGL-1, CD43, and CD44 are also known to associate with lipid rafts [158]. Further investigation into the effects of acidic lipids and cholesterol-enriched domains on the behaviors of these transmembrane proteins is warranted, especially considering that these molecules alter HIV-1 spread significantly when incorporated into virions (see below).

## 9. Roles Played by PSGL-1, CD43, and CD44 in Virus Spread

PSGL-1, CD43, and CD44 are type-1 transmembrane proteins that localize to the uropod in polarized T cells [159,160,161,162,163]. While these proteins play a role in T cell migration via interactions with selectin proteins expressed by endothelial cells [164,165], PSGL-1 and CD43 can serve as a physical barrier that inhibits cell-cell interactions because the extracellular domains of PSGL-1 and CD43 are highly glycosylated with sialylated *O*-linked glycans and thereby, highly extended [164,165,166,167,168,169,170]. CD44 is a major receptor for hyaluronan (HA), a glycosaminoglycan, and the interactions between CD44 and HA promote cell adhesion and cell-cell contacts [171].

It has been long known that CD44 is incorporated into HIV-1 [138,172,173]. Interestingly, virion-incorporated CD44 promotes trans-infection mediated by secondary lymphoid organ stromal cells, known as fibroblastic reticular cells (FRC). HA bound to virion-incorporated CD44 interacts with CD44 on the surface of FRCs and thereby promotes virus binding to FRCs (virus capture) and subsequent trans-infection of CD4^+^ T cells [174]. Of note, HA present on the surface of CD44-expressing CD4^+^ T cells prevents direct binding and infection of CD44-containing HIV-1 [175]. Therefore, CD44 incorporation into virus particles has a pro-viral effect on virus transmission to new host T cells in the presence of lymphoid organ stromal cells.

In contrast to CD44, PSGL-1 and CD43 reduce the infectivity of progeny HIV-1 when these proteins are expressed in virus-producing cells [176,177]. We and others have found that when PSGL-1 and CD43 are incorporated into virions, these proteins inhibit HIV-1 infection via blocking of virus attachment to target cells [178,179]. Interestingly, PSGL-1 and CD43 prevent Env-independent virus-cell binding, including virus capture by FRCs. Moreover, the intact extracellular domain of PSGL-1 is required for efficient suppression of virus attachment to target cells. Therefore, virion-incorporated PSGL-1 and CD43 likely inhibit virus-cell binding regardless of molecules mediating virus attachment, perhaps because of the extended extracellular domains. However, it is also possible that PSGL-1 and CD43 on the surface of HIV-1 particles attenuate not only virus-cell binding but also subsequent steps of the HIV life cycle, such as reverse transcription, as reported by another group [176]. Regardless of the mechanism, PSGL-1 and CD43 have antiviral effects on virus entry into new host cells even though they likely share with CD44 the mechanism of recruitment to virus assembly sites and hence, virion incorporation.

Altogether, as exemplified by the uropod-directed proteins discussed above, it is likely that the mechanism by which HIV-1 determines the sites of particle assembly has an impact on the composition of virus particles, thereby affecting the fate of the nascent virions either negatively or positively.

## 10. Conclusions and Future Directions

Assembly of HIV-1 Gag at the PM results in the formation of infectious progeny. The virion emerging from the host cell has an envelope that, besides Env glycoprotein, contains select lipid components and host proteins that are present at the PM. These host-derived components modulate the release, transmission, or entry of HIV-1 virions. Thus, the PM of virus-producing cells plays important roles not only as the virus exit sites but also as a platform for sorting components of virus particles.

It is well established that Gag-PI(4,5)P2 interaction mediates the specific localization of Gag to the PM. Cholesterol is another cellular lipid that plays an important role in Gag membrane binding. Interestingly, even though cellular PI(4,5)P2 by virtue of its sn-2 unsaturated acyl chain is unlikely to partition into lipid raft-like microdomains, the HIV-1 envelope is enriched in PI(4,5)P2, cholesterol, and lipid raft-like components compared to the producer cell PM. Incorporation of PM proteins into the HIV-1 envelope is likely to occur in various mechanisms based on the preference for raft-like microdomains, the membrane curvature associated with virus budding, and the potential enrichment of acidic lipids such as PI(4,5)P2. Interestingly, the proteins incorporated into the viral envelope could have positive (e.g., CD44) or the negative role (e.g., tetherin, CD43, and PSGL-1) in the HIV-1 life cycle.

Given the importance of PI(4,5)P2 in HIV-1 assembly and its potential role in virus spread via incorporation of host proteins, small molecule inhibitors that prevent Gag-PI(4,5)P2 interaction could be useful as inhibitors of HIV-1 assembly. Similarly, as a shift in the balance between MA-RNA and MA-PI(4,5)P2 binding could prevent Gag binding to the PM, the use of RNA aptamers that bind MA with higher affinity than PI(4,5)P2 might prove to be an effective antiviral strategy.

Several unanswered questions pertaining to HIV-1 assembly are evident from this review. What determines the preference of HIV-1 Gag for PI(4,5)P2 with at least one unsaturated acyl chain? How does PI(4,5)P2 outcompete the negative regulation of tRNA during Gag-membrane binding? Is there a difference among tRNAs in their ability to compete with acidic lipids? Where in the cytoplasm does a tRNA bind to Gag? Does RNA serve as a chaperone for other PM-targeting proteins in the cell? How does HIV-1 Gag reorganize the host membrane microdomains at the virus assembly sites? What molecule mediates the incorporation of PSGL-1, CD43, and CD44 into the HIV-1 envelope? How are the incorporated host proteins and lipids organized on the surface of virus particles? What is the collective impact of the proteins incorporated into the virus particle? Addressing these questions will inevitably advance our understanding of both cell biology of the PM and retrovirus assembly and spread.

## Figures and Tables

**Figure 1 viruses-12-00842-f001:**
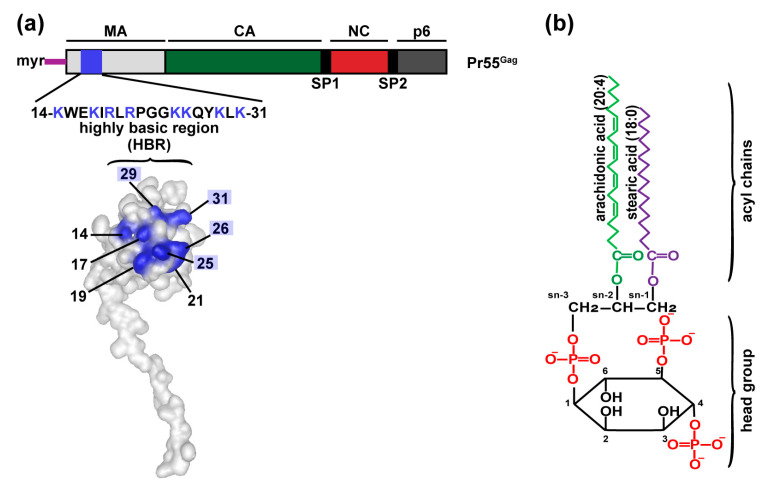
Cartoons depicting HIV-1 Gag MA-HBR, and PI(4,5)P2. (**a**) Schematic illustration of HIV-1 Gag with the structure of HIV-1 MA (PDB accession number 2HMX) showing the basic residues of the MA-HBR in blue. The key basic residues studied in detail in our recent work [48] are indicated (in the nomenclature for MA amino acid residues used in this review, the first methionine removed upon myristylation is not counted). (**b**) Illustration of 38:4 PI(4,5)P2, the predominant molecular species of PI(4,5)P2 in many primary mammalian cells [49,50,51,52,53].

**Figure 2 viruses-12-00842-f002:**
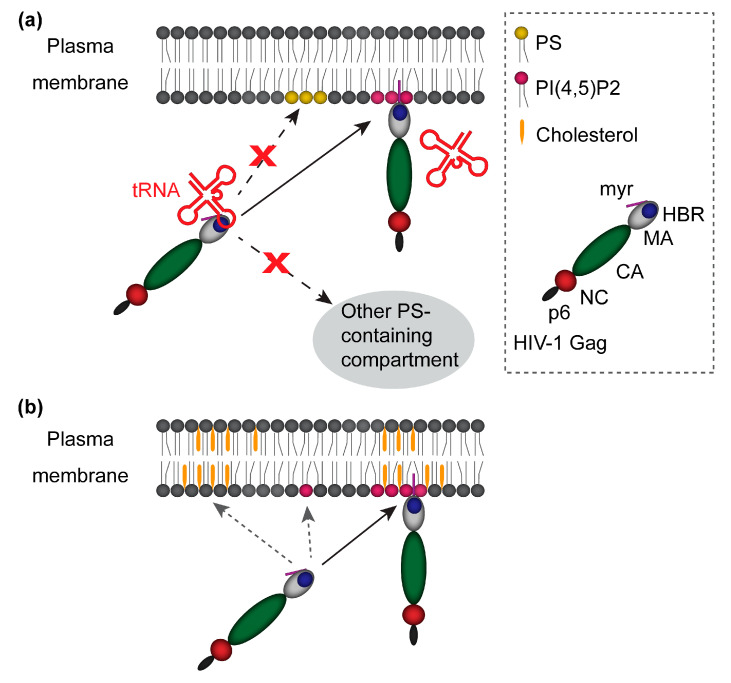
Regulation of Gag binding to the plasma membrane (PM). (**a**) Interactions of Gag with PI(4,5)P2 and tRNA ensure specific localization of Gag to the PM. Binding of Gag to tRNA prevents localization of Gag to intracellular membranes that contain PS but not PI(4,5)P2. Interaction of Gag with PI(4,5)P2-containing membrane overcomes the block on membrane binding by tRNA, thereby recruiting Gag to the PM; (**b**) cholesterol-dependent membrane binding by Gag. Cholesterol increases the membrane binding of Gag, including binding to PI(4,5)P2-containing PM.

**Figure 3 viruses-12-00842-f003:**
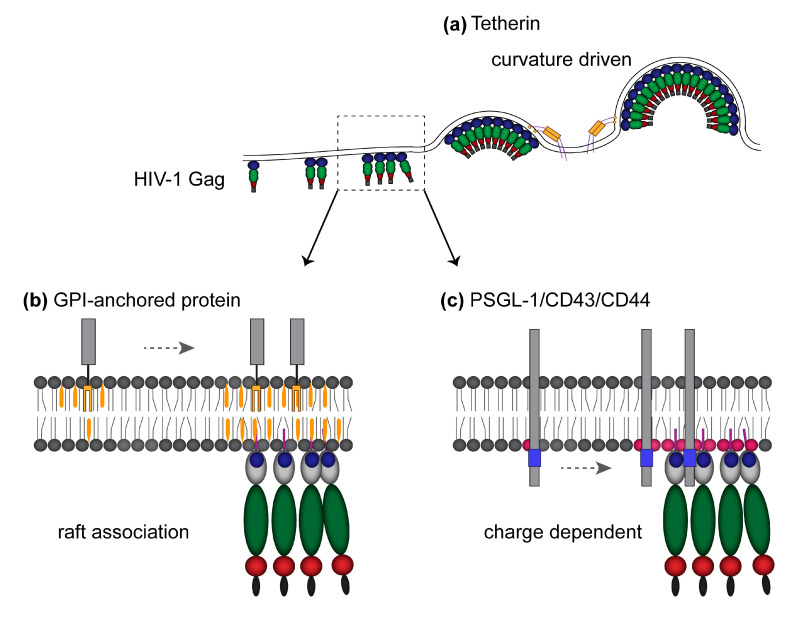
Mechanisms of host protein enrichment to HIV-1 assembly sites. (**a**) Membrane curvature promotes the incorporation of tetherin into nascent virions. (**b**) Preference of cellular proteins for raft-like domains promotes their enrichment at virus assembly sites. A GPI-anchored protein is shown as an example. (**c**) PSGL-1, CD43, and CD44, which contain a polybasic region in their cytoplasmic tails, cocluster with Gag in an HBR-dependent manner, potentially mediated by a polyacidic entity such as clusters of acidic lipids including PI(4,5)P2.

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
