# Peer review of "Rendezvous at Plasma Membrane: Cellular Lipids and tRNA Set up Sites of HIV-1 Particle Assembly and Incorporation of Host Transmembrane Proteins"

_viruses, 2020, doi:10.3390/v12080842_

Round 1

Reviewer 1 Report

The review by Thornhill et al provides  focused detailed information by compiling and examining the current understanding that directs HIV Gag to the plasma membrane and selection of host proteins into the virus particles.  The review provides insight into the role and interplay between of PI(4,5)P2 and tRNA in mediating specific localization of Gag to the plasma membrane.  Further details in the review includes the effects of PM lipids in virus assembly and incorporation of subset of transmembrane proteins driven by the membrane environment.  The mechanisms of HIV assembly at the plasma membrane as well as budding and release are complex.  Although we have gained significant knowledge and we still don’t have a complete picture.  The review brings out attention to the complexity of HIV particle assembly and raises several questions that needs to be addressed in the future.  This is a well written review and informative. 

Author Response

We thank the reviewer for the positive comments on our manuscript. The reviewer did not raise any concern or request for changes in the manuscript.

Reviewer 2 Report

Thornhill et al review the current knowledge of HIV assembly, specifically focusing on the early stages of plasma membrane targeting. This review provides a great background for virologists to understand plasma membrane lipids, namely phosphoinositide metabolism, to understand determinants required for Gag/MA targeting membranes and establishing a virus assembly site. The authors nicely highlight previous work and novel/new studies addressing the focus of the review. Further the authors don’t oversight their own works and balance in vitro and in cellulo studies and caveats regarding this system. I feel that this review will be a very nice contribution for the field of retrovirology and membrane enveloped viruses, therefore, I recommend accepting this review if the following major and minor points are addressed.

Major Points:

  • The authors describe the role of tRNAs in negatively regulating the association of Gag/MA with PS/PI(4,5)P2 membranes, can the authors comment in the section pertaining to in vitro conditions, whether the concentration of tRNAs used in these studies reflect the steady-state concentration of tRNAs that would be experienced by Gag/MA inside of cells? Perhaps a statement about the need to confirm these results with cellular studies interrogating the association of these moieties. While this reviewer appreciates the difficulties of interrogating RNA and lipid species in cellulo, this seems critical to advance our understanding of the interplay between RNA, PIPs, and Gag/MA.
  • Perhaps the authors could elaborate a bit more on potential mechanisms for the interplay between myristate exposure and tRNA binding to MA (Lines 213-214)?
  • The sentence for lines 227-229 could benefit from a few citations supporting the dynamic composition of the eukaryotic plasma membrane, namely the works of Kusumi and Simons.

Minor Points:

  • Line 73 needs a reference
  • Line 94, so many references for one finding, did all of these studies confirm this? Consider expanding to additional (or one more) sentence to describe the contribution of each study to this finding
  • Line 193, word choice: “same” -> ‘similar’ or ‘identical’
  • Line 194, ‘is’ missing ‘…where full length Gag is being produced’
  • Line 194, ‘rabbit reticulocyte lysates’ could be more clear using ‘in vitro transcribed’
  • Line 259, membranes plural
  • Section 6. A sentence could be added that supports the role of cholesterol in particle biogenesis based on enrichment in the virus (lipidomics studies previously cited).
  • Line 286, remove ‘the’ from end of line.
  • Line 339, consider replacing ‘trafficking’ with ‘migration’
  • In general use of articles in sentences could be improved

Author Response

We thank the reviewer for the positive comments and thoughtful suggestions on our manuscript. We provide below our point-by-point response below following the original comments by the reviewer.

---

Thornhill et al review the current knowledge of HIV assembly, specifically focusing on the early stages of plasma membrane targeting. This review provides a great background for virologists to understand plasma membrane lipids, namely phosphoinositide metabolism, to understand determinants required for Gag/MA targeting membranes and establishing a virus assembly site. The authors nicely highlight previous work and novel/new studies addressing the focus of the review. Further the authors don’t oversight their own works and balance in vitro and in cellulo studies and caveats regarding this system. I feel that this review will be a very nice contribution for the field of retrovirology and membrane enveloped viruses, therefore, I recommend accepting this review if the following major and minor points are addressed.

Major Points:

The authors describe the role of tRNAs in negatively regulating the association of Gag/MA with PS/PI(4,5)P2 membranes, can the authors comment in the section pertaining to in vitro conditions, whether the concentration of tRNAs used in these studies reflect the steady-state concentration of tRNAs that would be experienced by Gag/MA inside of cells? Perhaps a statement about the need to confirm these results with cellular studies interrogating the association of these moieties. While this reviewer appreciates the difficulties of interrogating RNA and lipid species in cellulo, this seems critical to advance our understanding of the interplay between RNA, PIPs, and Gag/MA.

[Response] We thank the reviewer for bringing up this point. We have added the information on the tRNA concentrations used in the in vitro studies, which are below or comparable to what would be experienced by Gag/MA inside of cells (lines 204-208). We further discussed limitations of the in vitro studies and highlighted the need of validations of the in vitro findings with in cellulo experiments (lines 211-220).

Perhaps the authors could elaborate a bit more on potential mechanisms for the interplay between myristate exposure and tRNA binding to MA (Lines 213-214)?

[Response] We added our speculation regarding the potential mechanisms for the interplay between myristate exposure and MA-tRNA binding (lines 231-235).

The sentence for lines 227-229 could benefit from a few citations supporting the dynamic composition of the eukaryotic plasma membrane, namely the works of Kusumi and Simons.

[Response] We thank the reviewer for the suggestion. We now cite several historically and/or conceptually important reviews from the Simons and Kusumi labs (line 250).

Minor Points:

Line 73 needs a reference

[Response] We now include the references for the statement in line 73.

Line 94, so many references for one finding, did all of these studies confirm this? Consider expanding to additional (or one more) sentence to describe the contribution of each study to this finding

[Response] We added a few sentences in this paragraph to highlight unique aspects of some of these studies (lines 95-103).

Line 193, word choice: “same” -> ‘similar’ or ‘identical’

Line 194, ‘is’ missing ‘...where full length Gag is being produced’

Line 194, ‘rabbit reticulocyte lysates’ could be more clear using ‘in vitro transcribed’

Line 259, membranes plural

Section 6. A sentence could be added that supports the role of cholesterol in particle biogenesis based on enrichment in the virus (lipidomics studies previously cited).
Line 286, remove ‘the’ from end of line.
Line 339, consider replacing ‘trafficking’ with ‘migration’

In general use of articles in sentences could be improved

[Response] We have made suggested changes above, except for the addition to Section 6 of a sentence about cholesterol enrichment in virus. This is because we already have a sentence to this effect in the beginning of Section 7 (lines 278-280). We feel it would be repetitive if we include the suggested sentence in Section 6. Instead, we have clarified this point in lines 289-290.